# Wearable Smart Fabric Based on Hybrid E-Fiber Sensor for Real-Time Finger Motion Detection

**DOI:** 10.3390/polym15132934

**Published:** 2023-07-03

**Authors:** Erhan Zhuo, Ziwen Wang, Xiaochen Chen, Junhao Zou, Yuan Fang, Jiekai Zhuo, Yicheng Li, Jun Zhang, Zidan Gong

**Affiliations:** 1Sino-German College of Intelligent Manufacturing, Shenzhen Technology University, Shenzhen 518118, China; jagerzhuo@foxmail.com (E.Z.); 2070412024@email.szu.edu.cn (Z.W.); chenxc506@foxmail.com (X.C.); zoujunhao00@gmail.com (J.Z.); fangoliver453@gmail.com (Y.F.); jakiezhuo@foxmail.com (J.Z.); yicheng.li@siemens-healthineers.com (Y.L.); 2Laboratory for Artificial Intelligence in Design, School of Fashion and Textiles, The Hong Kong Polytechnic University, Hong Kong 999077, China; alice.zj.zhang@connect.polyu.hk

**Keywords:** E-fiber sensor, polydimethylsiloxane, multiwalled carbon nanotube, smart fabric, finger motion detection

## Abstract

Wearable electronic sensors have attracted considerable interest in hand motion monitoring because of their small size, flexibility, and biocompatibility. However, the range of motion and sensitivity of many sensors are inadequate for complex and precise finger motion capture. Here, organic and inorganic materials were incorporated to fabricate a hybrid electronic sensor and optimized and woven into fabric for hand motion detection. The sensor was made from flexible porous polydimethylsiloxane (PDMS) filled with multiwalled carbon nanotubes (MWCNTs). The weight ratios of MWCNTs and geometric characteristics were optimized to improve the hybrid electronic sensor, which showed a high elongation at the breaking point (i.e., more than 100%) and a good sensitivity of 1.44. The strain-related deformation of the PDMS/MWCNT composite network resulted in a variation in the sensor resistance; thus, the strain level that corresponds to different finger motions is be calculated. Finally, the fabricated and optimized electronic sensor in filiform structure with a 6% MWCNT ratio was integrated with smart fabric to create a finger sleeve for real-time motion capture. In conclusion, a novel hybrid E-fiber sensor based on PDMS and MWCNTs was successfully fabricated in the current study with an optimal M/P ratio and structure, and textile techniques were adopted as new packaging approaches for such soft electronic sensors to create smart fabric for wearable and precise detection with highly enhanced sensing performance. The successful results in the current study demonstrate the great potential of such hybrid soft sensors in smart wearable healthcare management, including motion detection.

## 1. Introduction

Wearable devices can be worn on the human body as attachments, integrated with cloth and garments, or designed as fashion accessories. These have been developed and used for decades, ranging from wristwear, such as smart watches and fitness wristbands, to headwear, such as intelligent glasses and virtual reality headsets [1]. Such wearable electronics play an important role in the development of personalized healthcare systems that track key health indicators, for instance, the heart rhythm [2], blood pressure [3], body temperature [4], respiration rate [5], and body motion [6], for disease diagnosis, prevention, and intervention [7]. In this context, flexible or stretchable wearable systems that can be integrated flexibly with textiles for physiological detection are in growing demand. Meanwhile, flexibility and stretchability are also required to offer robust and conformal contact with the curved surfaces of the human body and to prevent the functionality of the devices from being affected by our daily activities.

Soft and wearable sensors with built-in electronic functions have recently gained considerable interest owing to their features of light weight, deformability, breathability, flexibility, comfort, etc., and they have been widely applied in healthcare management, including rehabilitation monitoring [8]. Zhang et al. [9] developed a flexible pressure sensor by utilizing a single layer composed of polystyrene microsphere arrays on polydimethylsiloxane (PDMS) to detect the human neck pulse. Mironov et al. [10] successfully fabricated a series of similar composite cylinders with a network of piezoelectric film sensors to achieve health monitoring. For people with functional impairments in their upper limbs, tracking their dynamic hand motions in rehabilitation treatment is crucial, given that hands are dexterous and account for about 90% of upper limb function [11,12]. To achieve precise hand motion monitoring, a variety of soft wearable sensors and devices have been designed and developed. These are usually nanocomposites that consist of elastic polymers to provide flexible backbones and metallic or carbonaceous nanoparticles for electrical characteristics. Nanoparticles include gold and silver nanowires [13], graphene [14], carbon black [15], and carbon nanotubes (CNTs), which are commonly used to improve the electrical performance of flexible sensors. Thus, E-signals can be produced in response to external mechanical inputs [16,17]. Three types of sensors, including capacitive [18], piezoelectric [19], and piezoresistive [20] types, are typically integrated with polymer materials for a variety of applications [21,22,23,24], among which flexible piezoresistive sensors with features of low power consumption, broad detection range, and easy manufacturing technique have been developed rapidly and become crucial candidates for wearable motion detection [17]. Many researchers have focused on such materials in the pursuit of high-performance flexible piezoresistive sensors. For instance, Li et al. [25] proposed a stretchable piezoresistive sensor made from ethylene propylene rubber with a silver conductive paint cover, which could quantitatively detect the simultaneous finger joint motion of dysfunctional hands in clinical applications. Additionally, Qu et al. [26] developed an ionic liquid hydrogel-based piezoresistive strain sensor with silver nanowires and a PAA network for real-time motion capture and gesture identification. Although advances in hand motion monitoring have been achieved, limitations persist, such as limited detection range, stability, and sensitivity, which need to be addressed to improve the accuracy performance of these devices in practical applications.

Considering the curved surfaces of the human body with hard or soft features, excellent elasticity and mechanical compliance are required for such sensors in wearable applications to adapt to mechanical bending, stretching, and twisting in some daily repetitive activities [27,28]. Fabric, which is interlaced with fibers and yarns to be soft and elastic, has been developed over thousands of years, with the original purpose of covering body and keeping it warm. Such fabrics are now integrated with advanced electronic science and technology to be an ideal wearable modality to fulfil the functions of monitoring, communication, therapy, assistance, and entertainment, among others [29]. Therefore, fibers with excellent elasticity and electrical function can provide excellent biocompatibility, permeability, and mechanical qualities for constructing ideal wearables and bio-interface electronics. The aforementioned polymer-based electronic sensors with excellent plasticity can be flexibly designed into filiform structure and used as yarns to create a new generation of smart fabric that can be directly worn on the soft and curved human body for a variety of applications. Zhang et al. [30] developed a core-spun structured smart yarn by wrapping nano silk fiber films toward rotating carbon nanotubes to obtain excellent electrical conductivity, strong mechanical strength, and great comfortability. Wang et al. [31] designed a fiber-shaped triboelectric nanogenerator composed of flame-retardant polyimide (PI) twisted conductive yarn. Such smart fiber can output a stable triboelectric signal via the interaction between the conductive core and PI shell, which acts as a self-powered sensor to provide the location information of wearers. Thus, fibers and fabric are an ideal wearable modality for the newly developed polymer-based sensors, which brings new inspiration to the structural design of wearable devices.

In this study, a novel wearable and flexible piezoresistive sensor based on a PDMS/multiwalled CNT (MWCNTS; P/M) nanocomposite is proposed for real-time hand motion detection. The conductive MWCNTs were used as inorganic filler materials and dispersed into the organic PDMS material, which has a low modulus, high elasticity, and excellent flexibility. A highly stretchable and flexible hybrid sensing nanocomposite was then prepared, and the device optimization was tested by adjusting the weight ratio of the inorganic and organic materials. To further enhance the sensing performance of the strain sensors, different structural designs, such as sheets and filiform, were systematically studied. Finally, the optimized filiform sensor—the hybrid E-fiber—was experimentally woven into a finger sleeve as smart fabric via textile techniques to realize the precise detection of finger and hand motions.

## 2. Materials and Methods

### 2.1. Materials and Equipment

The materials, including isopropyl alcohol with a spectral grade of over 99.5% (Alighting Chemical Reagent Network), PDMS and its curing agent (Shenzhen Osborn Co., Ltd., Shenzhen, China), MWCNTs with a purity of over 95%, a diameter between 8 to 15 nm, and a length of 50 um (Nanjing Xian Feng Nanomaterials Technology Co., Ltd., Nanjing, China), and polytetrafluoroethylene tubes (Shango Plastic Products Co., Ltd., Guangzhou, China) were prepared for hybrid sensor fabrication. Fabrication and performance characterization were evaluated using an ultrasonic cleaning machine (Xiamen Ultrasonic Instrument Co., Ltd., Xiamen, China), a magnetic mixer (Shanghai Kehuai Instrument Co., Ltd., Shanghai, China), an electrothermal blowing dry box (Shanghai bluepard instruments Co., Ltd., Shanghai, China), an electronic universal tensile testing machine (Shenzhen Kexing Precision Instrument Co., Ltd., Shenzhen, China), field emission scanning electron microscopy (SEM, Sirion, FEI Oregon, America), and a 34461A six-digit semi-digital multimeter (Keysight Technologies, Beijing, China). Unless otherwise noted, all materials and equipment were used as received and according to the recommendations from manufacturers.

### 2.2. Fabrication of the Hybrid Strain Sensor

A series of hybrid strain sensors with different material proportions and physical shapes (i.e., sheet and filiform) were fabricated according to the main steps shown in Figure 1. First, 30 mL of isopropyl alcohol (IPA) and 10 g of PDMS were mixed as the base material for 1 h in an ultrasonic cleaning machine for ultrasonication, and MWCNTS (0.6 g) was added to 30 mL of IPA for dispersion. Then, the base material and dispersed MWCNT solutions were mixed for the preparation of the sensing P/M material. The specific MWCNT weight ratios were set at 5%, 6%, 7%, 8%, 9%, and 10%, respectively. The mixtures were then heated for 20 h in the electrothermal blowing dry box at 100 °C for evaporation before blending with the PDMS curing agent in a weight ratio of 15:1. After defoaming in the degassing chamber, the mixtures were poured in predesigned molds (i.e., sheet and filiform shapes), and curing was performed at 60 °C for 10 h. The sheet samples were 100 mm long, 20 mm wide, and 3 mm thick, whereas the filiform samples were 3 mm in diameter and 100 mm long. Finally, flexible strain sensors with predefined shapes were fabricated, and their sensing performance was explored for optimization and further application.

### 2.3. Characterization and Measurement

Stretchability refers to the capacity of a sensor to continue functioning while being stretched, which depends on the type or ratio of nanomaterials, the structure of the sensor, and the fabrication process. To investigate and compare the performance of each fabricated sensor, dynamic compression tests were performed on an electronic universal tensile testing machine with a 100 N load cell. The sensors with different MWCNT weight ratios (5%, 6%, 7%, 8%, 9%, and 10%) were cyclically compressed for 10 cycles at various strain levels at a constant strain rate of 10 mm/s. Mechanical properties, including stress–strain curves, i.e., stretchability, were determined from the test. Additionally, the morphologies of the pure PDMS and flexible P/M nanocomposites were characterized by SEM.

Electrical performance under deformation of the hybrid sensor must be tested while applying finger motion detection. The stress–strain tensile properties of the hybrid sensors were tested by the electronic universal tensile testing machine at a tensile rate of 10 mm/min and a gauge length of 20 mm, and the piezoresistive sensors were placed between two copper plates electrically connected to a 34461A six-digit semi-digital multimeter to measure the resistance of the hybrid sensor throughout each dynamic compression.

Range of motion and sensitivity are the two dimensions for evaluating a soft sensor. The former is equal to the strain of the sensor, and the latter is equal to the resistance of the sensor. The ratio of the relative electrical signal change (often resistance or capacitance) to the applied strain is defined as GF. The strain sensitivity of the hybrid sensors is evaluated by GF [32]:(1)$$GF=\frac{R-R_{0}}{\varepsilon R_{0}}$$
where R represents the measured resistance, R_0_ is the initial resistance, and ε is the current strain. The GF of the sensor increases when the measured resistance increases faster than the current strain.

To detect and output human finger motions, the optimized sensor is woven into a fabric acting as the E-fiber to create a smart finger sleeve. The output voltage variation along with the finger motions was analyzed using the 34461A six-digit semi-digital multimeter. The usability and feasibility of the sensor for wearable equipment was demonstrated by measuring with the finger at various bending angles.

## 3. Result and Discussion

### 3.1. Stretchability of the Hybrid Sensors

The sheet and filiform structured P/M hybrid sensor samples were prepared for a stretchability test, as illustrated in Figure 2a,b. The stress–strain diagrams for both structured sensors with a range of MWCNT weight ratios are presented in Figure 2c,d, respectively. The ratio of 6% MWCNT offers a relatively higher stretchability for both structured sensors, i.e., the elongation at the breaking point for the sheet and filiform structure reached 107.89% and 113.044%, respectively. Additionally, three sets of repeatability experiments were conducted using a tensile machine with both structured sensor samples with MWCNTs of different weight ratios. The testing results as well as the standard deviation information are illustrated in Figure 2e. The tensile performance of the filiform structure sensors was generally better than that of the sheet sensors.

SEM images were taken to visually investigate the structure of the sensor at a 6% MWCNT ratio, particularly in terms of the variation in porosity and the quality of MWCNT dispersion within the matrix, as well as determining whether MWCNTs were successfully embedded into the polymer. First, the microstructures of the nanocomposites with varying levels of porosity were imaged, and the results of the sheet and filiform structured sensors at a 6% MWCNT ratio are shown in Figure 3a,b. The surface of the P/M nanocomposite was uneven when the conductive filler of the MWCNTs was uniformly dispersed into the PDMS matrix without agglomeration, and the pores of the sensor with filiform were thinner than that of the sheet sensor. As expected, a sensor with high stretchability displayed tiny pores in the microstructure, in contrast to the smooth cross-section of pure PDMS shown in Figure 3c. Evidently, the fillers were evenly distributed throughout the elastomeric matrix, forming an excellent conductive network when the filler loading reached its percolation threshold concentration of around 6% in the current study.

The SEM images of the sensors in the sheet and filiform structures differed, which may be due to the differences in the fabrication techniques that influence the dispersion and deposition of fillers among polymers. The sensor in the sheet structure was heated from the bottom, whereas the entire filiform structure was heated at the same time, since heat was transferred through the tube. Additionally, another nanocomposite-based sensor reported in a previous study, which was fabricated by heating from the bottom, had similar SEM images as our sheet sensors [33], indicating that fabrication techniques would result in different microstructures and properties of the sensors, although they have the same material proportion. In conclusion, sensors with a MWCNT weight ratio of 6% filiform structure may present a relatively higher stretchability in wearable applications.

### 3.2. Electrical Performance under Deformation

Electrical tests were performed on sheet P/M hybrid sensors with MWCNT weight ratios of 5%, 6%, 7%, 8%, 9%, and 10% to study their electrical performance under deformation, as presented in Figure 4a. The initial resistance (at 0% strain) of the samples decreased along with the increasing MWCNT mass fraction (Figure 4b). The results illustrated in the resistance–strain diagrams in Figure 4c–i indicate that the sheet structured sensor with 6% MWCNT ratio presented the highest resistance change at the same level of strain. Taking the maximum strain level of 25% as an example, the resistance variation in the hybrid sheet sensor with a 6% MWCNT weight ratio is at least twice the value of the sensor with a 5% MWCNT weight ratio, and up to four to five times higher than those of other proportions. Thus, a 6% MWCNT weight ratio of the P/M hybrid sheet sensor is an optimal proportion, offering excellent electrical performance.

Similarly, a cyclic resistance–strain variation experiment was conducted with filiform sensors with different MWCNT weight ratios. The results illustrated in Figure 5 are comparable to those obtained from the aforementioned sheet sensors. The resistance–strain diagrams in Figure 5b–h indicate that the filiform structured sensor with a 6% MWCNT ratio presented the highest resistance change at the same level of strain, followed by the ratio of 5%. Then, the variation significantly declined when the MWCNT weight ratio was over 6%.

When the MWCNTs disperse into the PDMS, electrons can pass through adjacent nanoparticles that are overlapping or in close distance. Strain-induced mechanical deformation would result in a disconnection between the fillers, thereby increasing the resistance of the hybrid sensors, as illustrated in Figure 4 and Figure 5. If there is a high loading of conductive fillers, it is more difficult to alter the conductive networks, and this can also degrade the stretchability of composites as well as the processability. A low concentration of fillers usually means a long distance between adjacent nanoparticles, which makes it difficult for electrons to pass through, corresponding to poor electrical performance. Considering the electrical performance under the deformation of both structured sensors, the ratio of conductive filler at 6% could be defined as the scale with largest sensing range and highest sensitivity, which offers a balance of stretchability and electrical conductivity. The experimental results showed that under the same experimental conditions and with the same MWCNT weight ratio of 6%, the filiform sensor exhibited a better electrical performance than the sheet structured sensor. This may be due to the denser microporous structure of the filiform sensor, which enhances its extendibility to obtain a greater mechanical deformation and resistance variation. This is consistent with the results reported in previous study conducted by Herren et al. [34].

### 3.3. E-Fiber-Based Smart Fabric for Finger Motion Detection

The fabricated and optimized E-fiber sensor with a 6% MWCNT ratio was finally applied to smart fabric to create a finger sleeve for motion capture. A series of textile techniques, including embroidering, knitting, and weaving, can be adopted to integrate the fiber sensor into smart fabric, as presented in Figure 6a–c. To investigate the sensor response to bending deformation, a plain weaving technique was used to fabricate the finger sleeve in a plan configuration with warp yarns and weft yarns interlaced one-by-one, forming a basic woven structure, as shown in Figure 6d. E-fibers were used as the warp yarns in the monolayer woven structure that followed the finger direction, attaching to each finger joint for finger motion detection. Meanwhile, traditional cotton fibers were used as the weft yarns to provide a soft and skin-friendly feeling with excellent flexibility and air permeability. Finally, the planar structured textile was sewn together in the form of a finger sleeve that conformed to the shape of the index finger and was subsequently placed on its joint, as shown in Figure 6e.

For the woven structured smart sleeve, sensitivity, defined as GF, was measured to be 1.3101, 0.2335, and 1.4405, respectively, within the corresponding strain regions of 0–21%, 21–33%, and 33–61%, as presented in Figure 7. The special GF variation behavior can be explained from a micro perspective. As the E-fiber was scanned to have a microporous structure, the nanoparticles of MWCNTs would be evenly distributed around the pores, such that each of the pores could be regarded as a conductive cell. When the breakdown of the initial brittle synergistic P/M conductive network is distributed in the outer layer of the cell skeleton upon tensile strain, this leads to a high GF value (GF = 1.3101) in a relatively tiny strain region I (0 < ε < 21%). Compared with the first stage, the damage effect of the second stage was weakened within the strain region II of 21% < ε < 33%, and the GF then decreased to 0.2335. When the strain increased to the region III (33% < ε < 61%), the breakdown of the initially brittle synergistic P/M conductive network distributed in the interior of the cell skeleton occurred and gradually suppressed the contact effect, leading to a higher GF of 1.4405. The GF variation phenomenon is quite similar to that reported in a previous study conducted by He et al., which could also be explainable from a microscopic view [35]. Additionally, the mechanical and sensing properties of a smart fabric can be influenced by other factors, such as the type of thread used (regular or elastic; thick or thin) and the fabric base frame (plain, twill, or sateen weaving, or other textile techniques). Thus, the incorporation of the developed E-fibers into different fabric structural designs resulted in varied sensing performance of the smart device, which needs to be scientifically considered in practical use.

In finger motion detection applications when wearing the E-fiber-based smart finger sleeve, the relative resistance variation in the sensor towards different finger joint bending angles was measured. The finger was first placed in a naturally straight position (0°) and bent to 30°, then returned to the initial position. This was repeated multiple times to record the corresponding resistance variation in real time. The bending test was conducted from 30° to 90° with 15° increments, and the results are presented in Figure 8a, indicating a good linear relationship between finger joint angles and resistance change. Angle–resistance diagrams of 30°, 45°, and 90° are demonstrated as examples in Figure 8b,c to show the stability of the E-fiber-based smart sleeve. Overall, the E-fiber sensor has great potential for human movement monitoring and can be flexibly integrated with wearable fabric for real-time detection applications.

To further demonstrate the stability of the sensor, 5000 cyclic experiments were performed with the maximum resistance in a cycle recorded every 500 intervals, as presented in Figure 9. A small resistance variation in the sensor was observed under different bending angles, demonstrating the feasibility and stability of this smart fabric in long-term practical use.

## 4. Conclusions

In summary, a novel hybrid E-fiber sensor based on PDMS and MWCNTs was successfully fabricated in the current study with an optimal M/P ratio of 6 wt%, and textile techniques were adopted as new packaging approaches for such soft electronic sensors to create smart fabric for wearable and precise detection with highly enhanced sensing performance. Specifically, the evenly distributed fillers among the micropores in a polymer base contributed to the unique properties of the sensors, including a remarkable sensitivity of 1.4405 and excellent stretchability (elongation at breaking point reached 113.044%). Even after subjecting the sensor to 5000 cycles of testing, the amplitude of the current variation remained consistent, demonstrating its excellent robustness and reliability. Notably, the GF values of the sensing were 1.3101, 0.2335, and 1.4405 in the three strain regions: 0–21%, 21–33%, and 33–61%, which is particularly promising for applications in wearable motion detection that require a large tensile range. Based on its lightweight design and exceptional flexibility, such a sensor can be seamlessly integrated into clothing, enabling the capture of precise motions, such as finger and hand movements. These practical applications represent only a fraction of the vast potential of robust pressure sensors, which may revolutionize smart fabric applications, including artificial intelligence and mobile health monitoring.

## Figures and Tables

**Figure 1 polymers-15-02934-f001:**
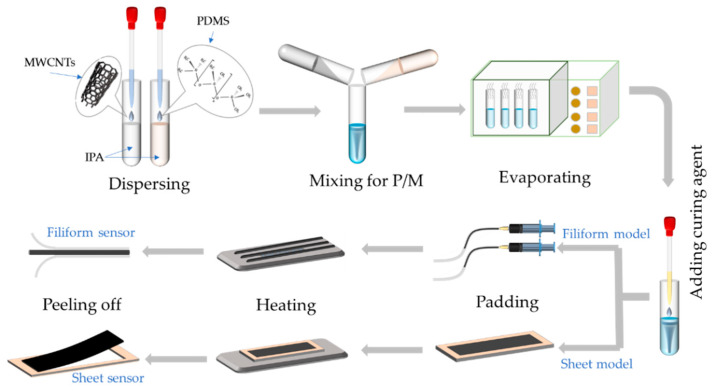
Schematics of fabrication processes of the hybrid strain sensors.

**Figure 2 polymers-15-02934-f002:**
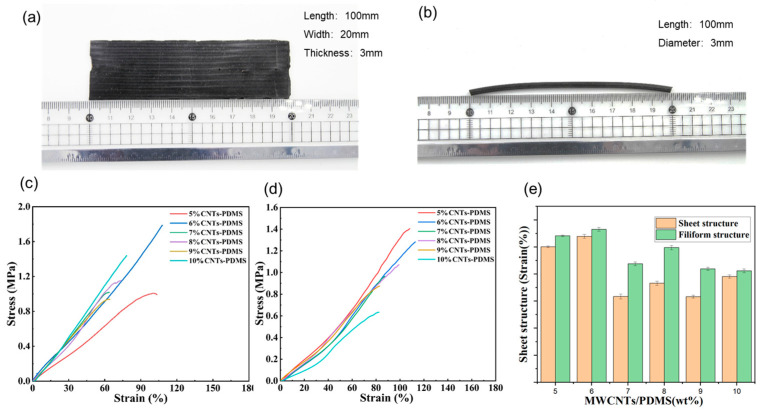
(**a**,**b**) Appearance of the sheet and filiform structured sensors; (**c**) stress–strain diagram of the sheet and (**d**) the filiform sensors; (**e**) yield strength comparations between the two structured sensors with various M/P ratios.

**Figure 3 polymers-15-02934-f003:**
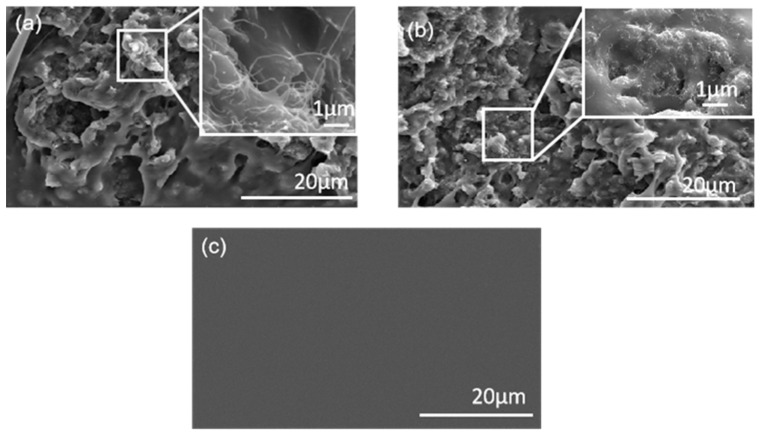
SEM cross-sectional morphology of (**a**) the sheet and (**b**) the filiform sensor at 6% M/P ratio; and (**c**) the pure PDMS structure.

**Figure 4 polymers-15-02934-f004:**
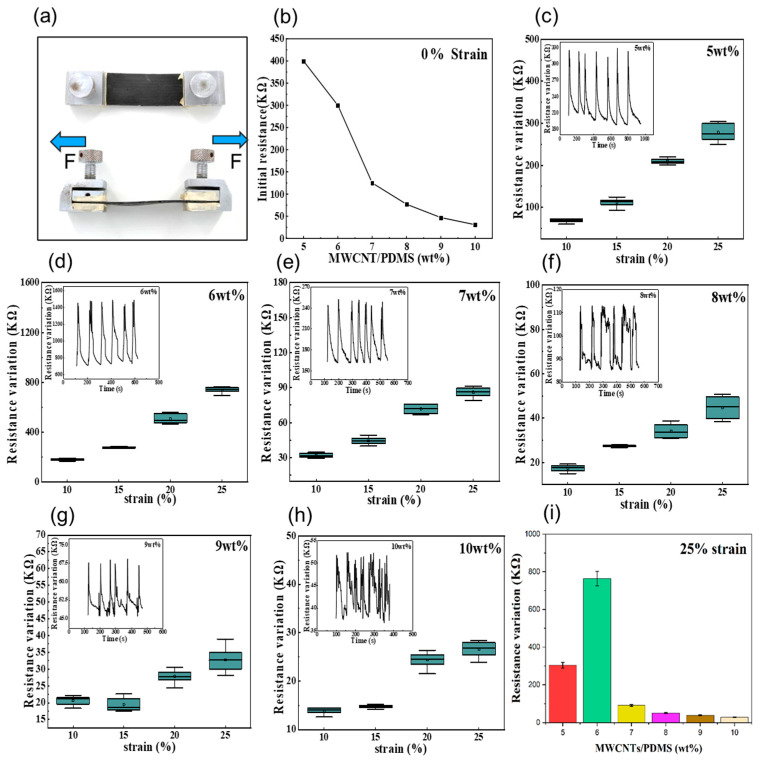
Electric characteristics of the flexible sheet sensor: (**a**) sheet sensor on test; (**b**) initial resistance variation with M/P weight ratio; (**c**–**h**) cyclic resistance–strain tests towards sheet sensors with different M/P weight ratio; (**i**) resistance changes of sensors of different mass fractions under the maximum strain of 25%.

**Figure 5 polymers-15-02934-f005:**
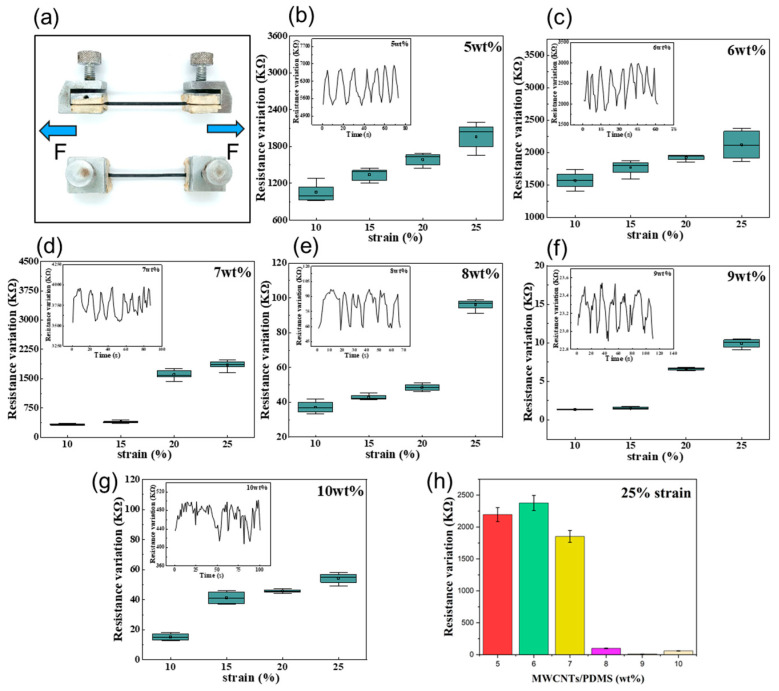
Electric characteristics of the flexible filiform sensor: (**a**) filiform sensor on test; (**b**–**g**) cyclic resistance–strain tests of sheet sensors with different M/P weight ratios; (**h**) resistance changes of P/M sensors of different mass fractions under the maximum strain of 25%.

**Figure 6 polymers-15-02934-f006:**
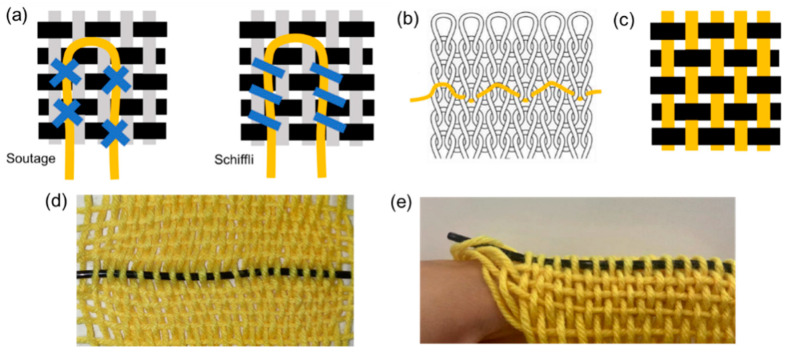
(**a**) Embroidering (i.e., soutage and schiffli); (**b**) knitting; (**c**) weaving; (**d**) smart fabric with E-fibers as the warp yarn; (**e**) the finger sleeve.

**Figure 7 polymers-15-02934-f007:**
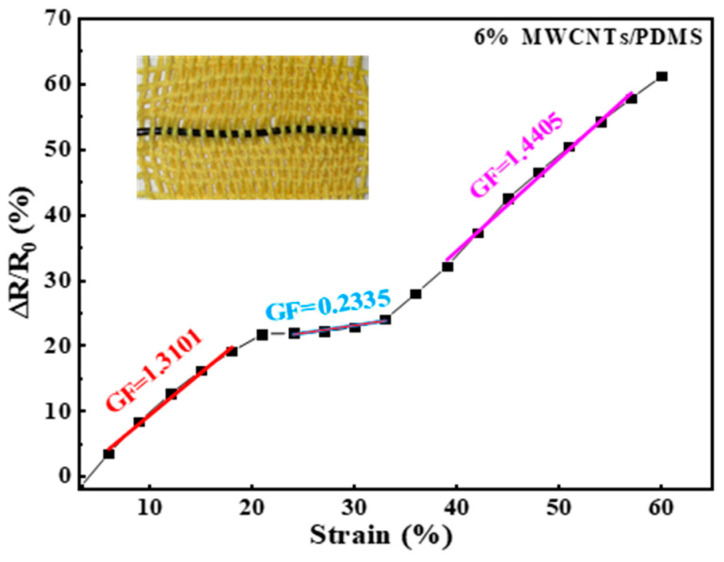
The GF sensitivity variation in the optimized E-fiber-based smart fabric.

**Figure 8 polymers-15-02934-f008:**
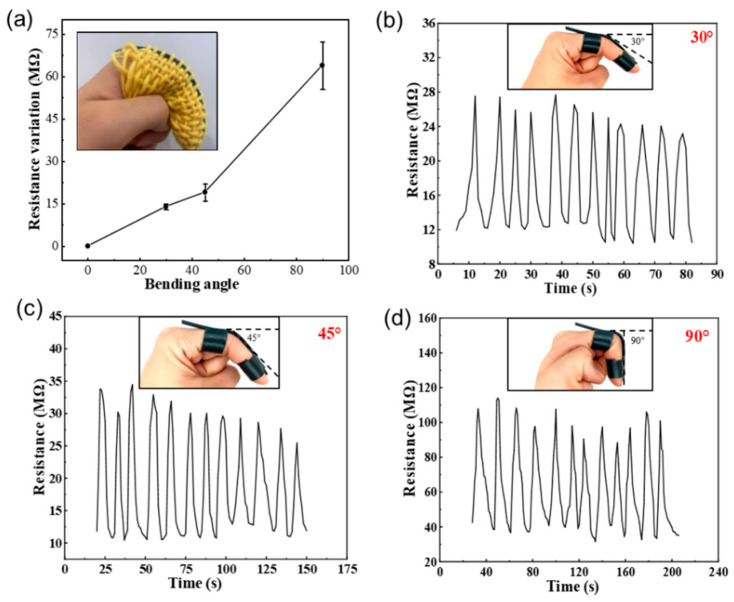
Application of the smart sleeve for finger motion detection: (**a**) the relationship between bending angle and resistance variation; (**b**–**d**) resistance variation in the sensor under different finger bending degrees of 30°, 45°, and 90°, respectively.

**Figure 9 polymers-15-02934-f009:**
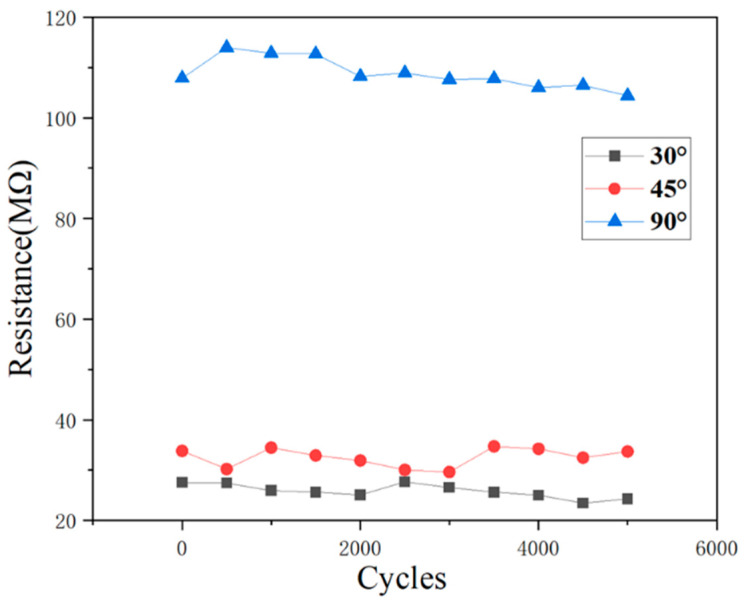
Resistance variation in the smart sensor in 5000 cyclic experiments under different bending angles.

## Data Availability

The data presented in this study are available on request from the corresponding author. The data are not publicly available due to privacy issues.

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
