# Peer review of "Wearable Smart Fabric Based on Hybrid E-Fiber Sensor for Real-Time Finger Motion Detection"

_polymers, 2023, doi:10.3390/polym15132934_

Round 1
Reviewer 1 Report
Manuscript 2478194 titled “Wearable Smart Fabric Based on Hybrid E-fiber Sensor for Real-Time Finger Motion Detection” summarizes fabrication and testing of PDMS/MWCNT composites films for wearable hybrid sensor applications. The manuscript summarizes nicely experimental results but is somewhat lacking on innovation. I recommend accepting for publication after addressing the comments below.
Comments:
Abstract/Conclusion: The authors should discuss aspects of innovation/new approach/novelty about their work.
Figure 2 (e) shows the yield strength comparison between the sheet and filiform structure samples as function of MWCNT wt%. The authors should include standard deviation information on the yield strength measurements by testing multiple samples. As presented, the 6 wt% MWCNT filiform sample has slightly better performance than the 6 wt% MWCNT sheet sample. However, without standard deviation information it is impossible to conclude which film (sheet versus filiform and 5 or 6 wt% doping) has the highest yield strength.
Figure 3 (c): add scale bar to PDMS SEM image.
Figures 4,5: Please discuss the insets. Why do the sheet films show slower resistance change over the filiform films? Or were the inset data collected under different strain rates for the two types of films?
Figure 8(b)-(c). The photo inset seems be identical for 30° and 45° finger bending. Please revise.
Conclusion: “Even after subjecting the sensor to 5000 cy-294 cycles of testing, the amplitude of the current variation remained consistent, demonstrating 295 its excellent robustness and reliability.” Can the authors include the results of this test in a Supporting Information file?
Reviewer 2 Report
I read with great interest the manuscript "Wearable Smart Fabric Based on Hybrid E-fiber Sensor for Real-Time Finger Motion Detection" and I consider that it is a strong paper with enough merits to be accepted for publication.
I have identified only minor issues:
-How advantageous is for an economic point of view to prepare such sensors related to the maximum number of cycles reported by authors (5000);
-in Section 2.2. the authors claimed that they prepared the composites without any compatibilisation treatment or chemical modification of MWCTNs before incorporation in PDMS matrix. It is known that the main difficulties appear by phase separation and agglomeration of MWCTNs. So that, I consider that here the authors must give details regarding the uniform distribution of MWCTNs even at high concentrations.
I consider that this paper can be accepted after Minor revision in Polymers.
